# Adaptive representations of sound for automatic insect recognition

**Marius Faiß**[1,2], **Dan Stowell**[1,3]*

**1** Naturalis Biodiversity Center, Leiden, The Netherlands, **2** Leiden University, Leiden, The Netherlands, **3** Department of Cognitive Science and AI, Tilburg University, Tilburg, The Netherlands

\* dan.stowell@naturalis.nl

## Abstract

Insect population numbers and biodiversity have been rapidly declining with time, and monitoring these trends has become increasingly important for conservation measures to be effectively implemented. But monitoring methods are often invasive, time and resource intense, and prone to various biases. Many insect species produce characteristic sounds that can easily be detected and recorded without large cost or effort. Using deep learning methods, insect sounds from field recordings could be automatically detected and classified to monitor biodiversity and species distribution ranges. We implement this using recently published datasets of insect sounds (up to 66 species of Orthoptera and Cicadidae) and machine learning methods and evaluate their potential for acoustic insect monitoring. We compare the performance of the conventional spectrogram-based audio representation against LEAF, a new adaptive and waveform-based frontend. LEAF achieved better classification performance than the mel-spectrogram frontend by adapting its feature extraction parameters during training. This result is encouraging for future implementations of deep learning technology for automatic insect sound recognition, especially as larger datasets become available.

**Data Availability Statement:** Datasets are publicly available: InsectSet32 at https://zenodo.org/record/7072196 InsectSet47 & InsectSet66 at https://zenodo.org/record/8252141 Software source code: InsectSet32 at https://github.com/mariusfaiss/

## Author summary

Insects are crucial members of our ecosystems. These often small and evasive animals have a big impact on their surroundings, and there is widespread concern about possible population declines. However, it can be difficult to monitor them in sufficient detail. We investigated an under-used evidence stream for insect monitoring: their sounds. Combining recent advances in deep learning, with newly curated open datasets of insect sound, we were able to train machine learning systems to identify insect species with encouraging strong performance. Since insect sounds are very different from human sounds, a key part of our investigation was to compare a standard (spectrographic) representation of sound against an automatically-optimized representation called LEAF. Across three different datasets we found LEAF led to more reliable species recognition. Our work demonstrates that sound recognition can be effective as a new evidence stream for insect monitoring.

InsectSet32-Adaptive-Representations-of-Sound-for-Automatic-Insect-Recognition InsectSet47 & InsectSet66 at https://github.com/mariusfaiss/InsectSet47-InsectSet66-Adaptive-Representations-of-Sound-for-Automatic-Insect-Recognition.

**Funding:** MF was supported by a Martin &Temminck Fellowship (Naturalis Biodiversity Center), which provided him with a salary. The funders had no role in study design, data collection and analysis, decision to publish, or preparation of the manuscript.

**Competing interests:** I have read the journal's policy and the authors of this manuscript have the following competing interests: DS is an Academic Editor for PLOS Computational Biology.

## Introduction

The insect order Orthoptera forms the animal clade with the most species capable of acoustic communication, with about 16,000 species using acoustic signals for sexual communication, and even more species displaying acoustic defensive signaling [1]. The main mode of sound production in Orthoptera is stridulation, where body parts are rubbed against each other to create audible vibrations, with one body part having a row of fine teeth and the other being equipped with a plectrum that sets the teeth into vibration. Most of the 3200 species in the family Cicadidae produce sound by rapidly deforming tymbal membranes, producing series of loud clicking sounds that set the tymbals into resonance [2–4]. Many of these sounds are species-specific, and in some cases are key criteria for species identification [5].

Declines in insect population numbers have been receiving wide attention in the scientific community as well as the public, but many of these reports only sample a small number of representative species or focus on limited geographic locations [6,7]. To implement effective conservation efforts, populations need to be monitored more closely and widely across species and geographic locations [6]. Insects, and specifically Orthoptera and Cidada, are a difficult group to detect with conventional monitoring methods such as visual surveys and various trapping strategies [8]. This is mainly due to their small size, camouflage and cryptic lifestyles in often inaccessible and difficult environments such as tropical rainforests [9]. Such species might be detected much more easily by the sounds they produce. Acoustic monitoring methods focused on Orthoptera have been successfully used for detection of presence and absence of species, determining distribution ranges, detection of otherwise cryptic species [10] and evaluating quality and deterioration of habitats, since they can function as indicator species [11]. Additionally, this method is mostly non-invasive, less elaborate than other common monitoring approaches [8] and could be automated to a high degree [9]. Video monitoring in comparison, is highly dependent on lighting conditions and direct visual contact with the subjects, and consumes more energy as well as data storage [12].

In the present work, we develop a robust method for acoustic classification of orthopteran and cicada species, using a deep learning method that can adapt to acoustic characteristics of the targeted insects. Some previous attempts of identifying Orthoptera by their sounds have focused on using manual extraction of sound features such as carrier frequency or pulse rates [10,13]. These features must be manually selected and their parameters defined before use for automatic classification. Selected features and parameters might not perform well in all situations however, such as when background noise disturbs waveform feature measurements, when non-target species produce very similar sounds, or when target species show strong variation of certain parameters [14]. For example, ambient temperature during the recording can influence the frequency of Orthoptera song as a result of being poikilothermic organisms [15]. Orthoptera regulate their speed of muscular contraction with the ambient temperature during song production. This results in higher frequency sounds and especially increased pulse rates with higher temperatures in most Orthoptera [15,16]. Auxiliary factors such as temperature can be included as inputs to many classification algorithms; nevertheless, such variations complicate the task of feature extraction. Hao et al. [14] explored an alternative way to avoid manually specifying acoustic features, by defining a "textural" similarity measure between spectrogram patches, using a general compression-based distance calculation. But the approach of automatically optimizing the representation parameters to the input data is still unexplored to our knowledge, especially for use with neural networks.

Deep learning methods are a more recent promising approach for acoustic monitoring tasks, as they can classify complex acoustic signals with high accuracy and little to no manual pre-processing of the input data [17]. Combined with sound event detection (SED), long-form

field recordings can be classified without any manual extraction of features or relevant clips to be identified. There are however a number of challenges to overcome, some practical and some related to the specific species traits. For applying machine learning methods, large, diverse and balanced annotated datasets are needed to train and test the algorithms.

Before an audio recording can be fed into a neural network to be analyzed, the high-resolution waveform has to be reduced to a feature space that can be processed and interpreted by a neural network [18,19]. The common approach for audio classification tasks has historically been inspired by the human perception of frequency and loudness. This is in part due to the focus of many of the early audio classification tasks that were heavily researched: speech or language recognition, or music-based analysis tasks [19]. All the relevant acoustic information for these tasks is contained in and optimized for human auditory perception, or vice versa. Humans experience frequency and loudness on non-linear scales [20]. Linear changes in frequency towards the lower frequency spectrum generally sound more obvious, while the same difference in frequency applied to a higher register can be undetectable to the human ear. In compressing the spectral energy of a signal for analysis in a neural network, these characteristics of human perception are applied with the use of the so-called mel-filter banks.

First, the input audio waveform is transformed into a spectrogram using the short-time Fourier transform (STFT), dissecting the signal into pure sine-wave frequencies and their respective energies [18,20]. Then, the mel-filter banks are applied, consisting of triangular bandpass filters, spaced along a logarithmic scale over the sampled frequency spectrum. These filters pool the energy of all frequencies that lie within their range, using a windowing function. This reduces the resolution from a high sample rate down to a number of frequency bins that can be easily analyzed. Following this, loudness compression is applied, also based on the non-linearity of human hearing [18], resulting in a mel-spectrogram, that can essentially be treated like an image by a neural network. These processing methods, especially the filter banks, rely on hand-crafted parameters that may not relate in any way to the sounds to be analyzed in a specific task. The logarithmic frequency scaling for example results in high spectral resolution in lower frequency ranges, but groups together larger and larger frequency ranges in higher registers, thereby potentially obscuring relevant high-frequency information and focusing on lower frequency bands when they do not necessarily contain relevant information (Fig 1).

Insect sounds are not generated using a source-filter mechanism as in mammals or birds, but with stridulatory or tymbal mechanisms that create a different structure of frequencies and overtones [2–4,16,21,22]. The sounds are often non-harmonic, broadband buzzing and chirping sounds with amplitude modulations and up to several minutes in length, or much shorter clicking sounds of less than 1 ms [16,23]. Generally, insect sounds are much higher in frequency than most mammal or bird sounds, with many species producing ultrasonic sounds, some up to 150 kHz [2,23,24]. This emphasis on high-frequency sounds, sometimes entirely and far outside of the human hearing range (~20 Hz—20 kHz) could have an impact on the performance of audio classification networks, depending on their approach. It is likely that the mel-filter bank approach based on human perception is not optimal to recognize and discriminate between subtle differences in high frequencies for many insect sounds, even if it works well enough for other sounds such as birdsong. Despite this, many previous attempts of classifying Orthoptera and Cicada by their sounds have used various versions of mel spectrograms, sometimes in combination with other handcrafted features or spectrogram modifications [25–27].

Recent work in deep learning has introduced adaptive, waveform-based methods such as LEAF [18], replacing the predefined spectrogram calculation with a parametric transform whose parameters are optimized at the same time as the rest of the network. These could potentially optimize their extraction of audio features to better fit insect sounds. The LEAF

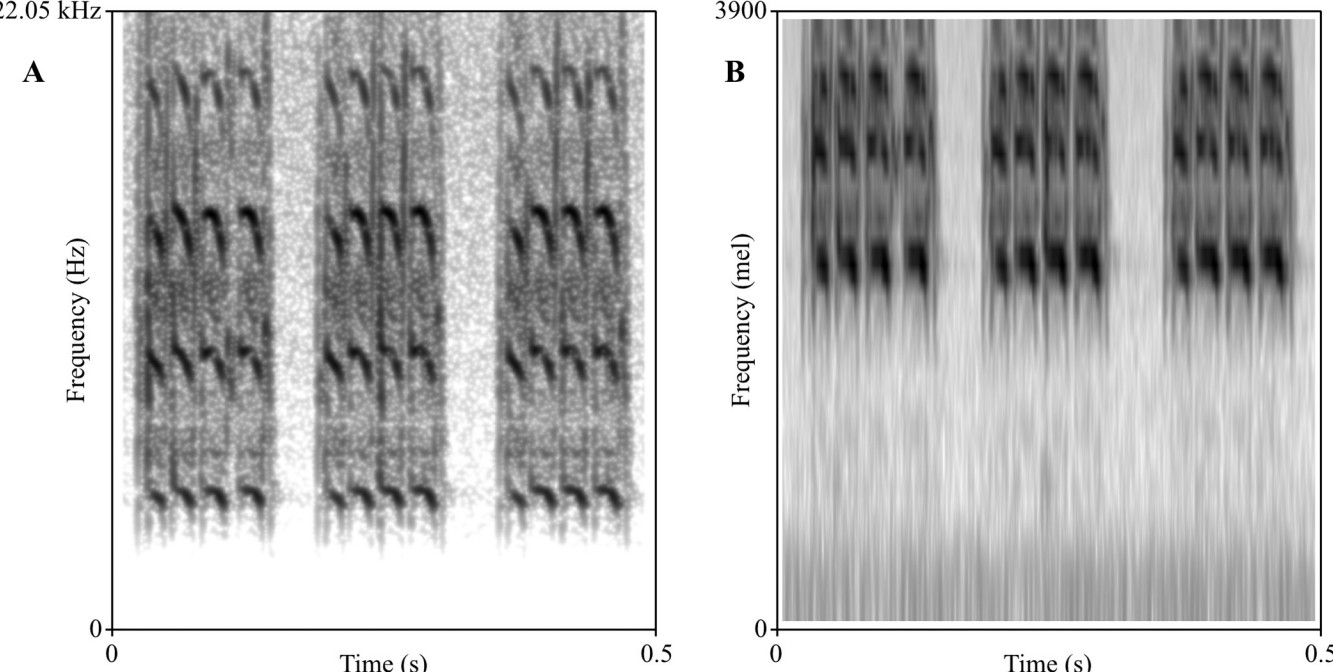

**Fig 1. Two spectrograms of the same recording of *Gryllus campestris*.** Spectrogram A displays the frequency axis linearly in Hz. Spectrogram B uses the mel frequency scale, which compresses the frequency axis to show higher resolution in lower frequency bands than in higher bands, mimicking the human perception of frequency. Both spectrograms display the same spectrum of frequencies. Due to the mostly high-frequency information and empty low frequencies in this recording, the mel spectrogram B obscures a large amount of information compared to the linear spectrogram A.

frontend allows the adjustment of filter frequency and bandwidth as well as normalization and time-pooling parameters during training to adapt to the data [18]. This frontend has been evaluated on a diverse set of audio classification tasks involving human-centric sound such as language, music, emotion, speaker recognition and more, and has shown improved performance over the standard Mel spectrogram approach in many cases [18]. But so far, it has not been evaluated on classification tasks involving sound sources that are less fit to the human perception of sound. For uses like insect species recognition that are much higher pitched and structured differently than human sounds, this frontend could be especially advantageous. It could adapt to the characteristics of insect sounds by learning increasing spectral resolution in higher frequency ranges, selecting and focusing on meaningful frequency bands that are otherwise pooled together, and learning how to ideally pool and compress these bands individually. Accordingly, the high resolution in lower frequency ranges that is present in Mel-filter bank approaches could be reduced or completely omitted, since it is rarely present in insect sounds [23].

The potential of deep learning methods for insect sound classification has not been studied extensively yet, especially their performance with adaptive frontends and extended sample rates/frequency ranges. In the present work, the performance of two different machine learning approaches will be tested in species classification of insect sound recordings, with only one species present at once. Complicating environmental conditions like distance from the recorder or background noise will be introduced by data augmentation methods to increase the diversity of the data set and improve the generalizability of the networks. The goal is to explore the potential for using deep learning methods to classify Orthoptera and Cicadidae with sounds recorded by entomologists and citizen scientists, and to evaluate the potential advantage of adaptive frontends for feature extraction of non-human, high-frequency sounds.

## Methods

We tested the performance of two audio feature extraction methods acting as frontends to a convolutional neural network. We compared the classic mel-spectrogram frontend to the adaptive and waveform-based frontend LEAF. It is initialized to function similarly to the mel frontend before training, but its parameters can be adjusted during training [18]. As a backend classifier, a convolutional neural network optimized for audio classification was implemented and adapted [28]. The frontends were tested on three increasingly large datasets of insect recordings.

### InsectSet32

Since larger collections of insect recordings have only recently become publicly available, the dataset used for initial tests ("InsectSet32") was compiled from private collections of Orthoptera and Cicadidae recordings (Orthoptera dataset by Baudewijn Odé and Cicadidae dataset by Ed Baker which includes recordings from the Global Cicada Sound Collection hosted on Bioacoustica [29], including recordings published in [30,31]). These recordings were conducted in the field as well as in laboratory settings on a variety of recording devices. Only files in WAV audio format with sample rates of 44.1 kHz or higher were included. All files were converted to mono and the sample rates were standardized to 44.1 kHz by down sampling recordings with higher resolutions. The files were manually auditioned to exclude files that contained strong noise interference, sounds of multiple species or other audio distortions and artifacts. Many recordings included voice over commentary at the beginning of the recordings. Only the last ten seconds of audio from these recordings were used, to automatically exclude the commentary. Only species with at least four usable audio recordings were included in the final dataset. Overall, 32 species were selected, with 335 files and a total recording length of 57 minutes and four seconds (Table 1). Between species, the number of files ranges from four to 22 files and the length from 40 seconds to almost nine minutes of audio material for a single species. The files range in length from less than one second to several minutes.

For training and evaluating the two frontends, InsectSet32 was split into the training, validation and test sets [12]. Due to the low number of files in some classes, the split into the three subsets was done for all classes individually to ensure that each class is represented in all three

**Table 1. InsectSet32: 335 files from 32 species with a total recording length of 57 minutes and four seconds were selected from two different source datasets (Orthoptera dataset by Baudewijn Odé and Cicadidae dataset by Ed Baker).** Number of files (n) and total length of recordings (min:s) per species.

| Baudewijn Odé—Orthoptera | | | Ed Baker—Cicadidae | | | | | |
|---|---|---|---|---|---|---|---|---|
| Species | n | min:s | Species | n | min:s | Species | n | min:s |
| Chorthippus biguttulus | 20 | 3:43 | Azanicada zuluensis | 4 | 0:40 | Platypleura divisa | 6 | 1:00 |
| Chorthippus brunneus | 13 | 2:15 | Brevisiana brevis | 5 | 0:50 | Platypleura haglundi | 5 | 0:50 |
| Gryllus campestris | 22 | 3:38 | Kikihia muta | 6 | 1:00 | Platypleura hirtipennis | 6 | 0:54 |
| Nemobius sylvestris | 18 | 8:54 | Myopsalta leona | 7 | 1:10 | Platypleura intercapedinis | 5 | 0:50 |
| Oecanthus pellucens | 14 | 4:27 | Myopsalta longicauda | 4 | 0:40 | Platypleura plumosa | 19 | 3:09 |
| Pholidoptera griseoaptera | 15 | 1:54 | Myopsalta mackinlayi | 7 | 1:08 | Platypleura sp04 | 8 | 1:20 |
| Pseudochorthippus parallelus | 17 | 2:01 | Myopsalta melanobasis | 5 | 0:43 | Platypleura sp10 | 16 | 2:24 |
| Roeseliana roeselii | 12 | 1:03 | Myopsalta xerograsidia | 6 | 1:00 | Platypleura sp11 cfhirtipennis | 4 | 0:40 |
| Tettigonia viridissima | 16 | 1:34 | Platypleura capensis | 6 | 1:00 | Platypleura sp12 cfhirtipennis | 10 | 1:40 |
| | | | Platypleura cfcatenata | 22 | 3:34 | Platypleura sp13 | 12 | 2:00 |
| | | | Platypleura chalybaea | 7 | 1:10 | Pycna semiclara | 9 | 1:30 |
| | | | Platypleura deusta | 9 | 1:23 | | | |

subsets (tr/val/te) and to prevent data leakage. The resulting split amounts to 62.7% of the files being used for training, 15.2% for validation and 22.1% for testing. The dataset is publicly available on zenodo.org [32].

## InsectSet47

After initial tests on InsectSet32 were conducted, a large collection of high-quality Orthoptera recordings by experts and citizen scientists was published on xeno-canto.org. From this collection, WAV files with sample rates of at least 44.1 kHz were downloaded and manually auditioned to compile a more diverse dataset together with the recordings from InsectSet32. Many recordings had been filtered or upsampled to 44.1 kHz by the uploaders, which was evident by a lack of audio information in certain frequency areas (most commonly above 16 kHz due to initially lower sample rates). Only full spectrum recordings were selected.

The files include sound snippets of single insect calls only seconds in length as well as long-term recordings of insect songs reaching up to 20 minutes. Many of the longer files included periods of silences without insect sounds. To exclude these silent periods, files that contained periods without insect sound of more than five seconds were edited into one or more files that contained only the insect sounds. The resulting edited snippets from one original recording were treated as one audio example to prevent them from ending up in multiple data sub-sets (train, test, validation) during the model training and evaluation process. Only species with at least ten usable recordings were included in the dataset. The recordings from the source datasets used for InsectSet32 (by Baudewijn Odé and Ed Baker) were also included in this selection process. Due to the more detailed editing process used for Dataset47, more audio material was gathered this time, but fewer species were included due to the higher minimum number of files per species. Therefore, InsectSet32 is only partially included in Insectset47. Overall, 47 species were selected for InsectSet47, with overall 1006 files and a total recording length of 22 hours (Table 2).

**Table 2. InsectSet47: 1006 files from 47 species with a total recording length of 22 hours were selected mainly from xeno-canto.org, as well as two private collections (Orthoptera dataset by Baudewijn Odé and Cicadidae dataset by Ed Baker).** Number of files (n) and total length of recordings (min:s) per species.

| Species | n | min:s | Species | n | min:s | Species | n | min:s |
|---|---|---|---|---|---|---|---|---|
| Chorthippus biguttulus | 52 | 29:49 | Acheta domesticus | 23 | 55:38 | Gomphocerus sibiricus | 14 | 26:04 |
| Stenobothrus stigmaticus | 39 | 5:31 | Oecanthus pellucens | 22 | 28:38 | Barbitistes yersini | 14 | 19:59 |
| Chorthippus mollis | 38 | 27:35 | Platypleura cf catenata | 22 | 17:46 | Pholidoptera aptera | 13 | 10:31 |
| Gryllus campestris | 38 | 94:21 | Omocestus rufipes | 21 | 16:28 | Pholidoptera littoralis | 13 | 4:00 |
| Conocephalus fuscus | 34 | 53:06 | Pholidoptera griseoaptera | 21 | 11:46 | Metrioptera brachyptera | 13 | 20:29 |
| Roeseliana roeselii | 33 | 33:39 | Chorthippus apricarius | 20 | 28:27 | Leptophyes punctatissima | 13 | 26:47 |
| Pseudochorthippus parallelus | 33 | 24:36 | Phaneroptera falcata | 20 | 28:29 | Pseudochorthippus montanus | 12 | 11:29 |
| Chorthippus brunneus | 32 | 20:58 | Myrmeleotettix maculatus | 20 | 55:06 | Platypleura sp13 | 12 | 7:01 |
| Tettigonia cantans | 32 | 57:15 | Platypleura plumosa | 19 | 14:41 | Chorthippus albomarginatus | 11 | 40:29 |
| Decticus verrucivorus | 31 | 71:30 | Stenobothrus lineatus | 18 | 32:41 | Eupholidoptera schmidti | 11 | 9:39 |
| Ephippiger diurnus | 29 | 39:33 | Conocephalus dorsalis | 18 | 23:07 | Melanogryllus desertus | 11 | 25:24 |
| Gomphocerippus rufus | 28 | 29:38 | Chrysochraon dispar | 17 | 15:35 | Tylopsis lilifolia | 11 | 3:30 |
| Nemobius sylvestris | 28 | 38:11 | Gryllus bimaculatus | 17 | 27:32 | Omocestus petraeus | 10 | 9:21 |
| Gampsocleis glabra | 26 | 55:01 | Platypleura sp10 | 17 | 17:55 | Chorthippus vagans | 10 | 11:43 |
| Omocestus viridulus | 25 | 45:25 | Phaneroptera nana | 16 | 29:53 | Platypleura sp12 cf hirtipennis | 10 | 7:41 |
| Tettigonia viridissima | 24 | 25:30 | Platycleis albopunctata | 15 | 24:44 | | | |

## InsectSet66

InsectSet47 was expanded to include even more species and audio examples with citizen scientist recordings from iNaturalist.org. More frequently than in the previous source collections, many recordings had been filtered, data-compressed or heavily edited, including time-stretching and pitch shifting. These files were not selected. Additionally, a substantial number of recordings were submitted multiple times as separate observations. These recordings were only included once in the final dataset, unless they were logged as multiple different species, in which case they were completely excluded. Otherwise, the same selection process as before was used and the dataset was expanded to include 66 species ("InsectSet66"), 1554 recordings and a total length of over 24 hours (Table 3). Between species, the number of files ranges from ten files and a minimum length of 80 seconds to 152 files and almost 98 minutes of audio material for a single species.

InsectSet47 and InsectSet66 were split into the training, validation and test sets while ensuring a roughly equal distribution of audio files and audio material for every species in all three datasets. To achieve this, files were sorted by file length for each species separately. They were then distributed into the three datasets by following a repeating pattern. The two longest files were moved into the training set, the third largest into the validation set, the fourth largest into the test set. The files at positions five and six were assigned to the training set again, the seventh largest to the validation set, the eighth to the test set. The ninth and tenth files were moved into the training set and the pattern was repeated for the remaining files if there were more than ten (1: train, 2: train, 3: val, 4: test, 5: train, 6: train, 7: val, 8: test, 9: train, 10: train, 11: repeat from 1). This resulted in a 60/20/20 split (train/validation/test) by file number and a 64/19.5/16.5 split by file length. InsectSet47 and InsectSet66 are publicly available on zenodo.org [33].

**Table 3. InsectSet66: 1554 files from 66 species with a total recording length of 24 hours and 32 minutes were selected from five different source datasets (Orthoptera and Cicadidae datasets from iNaturalist, Orthoptera dataset from xeno-canto, Orthoptera dataset by Baudewijn Odé and Cicadidae dataset by Ed Baker).** Number of files (n) and total length of recordings (h:min:s) per species.

| Species | n | h:min:s | Species | n | h:min:s | Species | n | h:min:s |
|---|---|---|---|---|---|---|---|---|
| Yoyetta celis | 152 | 0:11:16 | Aleeta curvicosta | 23 | 0:04:04 | Gomphocerus sibiricus | 14 | 0:26:05 |
| Gryllus campestris | 57 | 1:37:39 | Platypleura cfcatenata | 22 | 0:17:47 | Barbitistes yersini | 14 | 0:19:59 |
| Chorthippus biguttulus | 53 | 0:30:25 | Omocestus rufipes | 22 | 0:16:34 | Psaltoda plaga | 14 | 0:04:21 |
| Galanga labeculata | 43 | 0:06:16 | Chorthippus apricarius | 21 | 0:28:35 | Popplepsalta notialis | 14 | 0:02:58 |
| Yoyetta repetens | 40 | 0:05:23 | Myrmeleotettix maculatus | 21 | 1:05:37 | Pholidoptera littoralis | 13 | 0:04:00 |
| Chorthippus mollis | 39 | 0:27:50 | Cicada orni | 21 | 0:06:50 | Pseudochorthippus montanus | 13 | 0:11:36 |
| Stenobothrus stigmaticus | 39 | 0:05:31 | Phaneroptera falcata | 20 | 0:28:30 | Leptophyes punctatissima | 13 | 0:26:48 |
| Pseudochorthippus parallelus | 37 | 0:25:08 | Gryllus bimaculatus | 20 | 0:28:44 | Cyclochila australasiae | 13 | 0:01:53 |
| Roeseliana roeselii | 37 | 0:34:34 | Platypleura plumosa | 19 | 0:14:42 | Platypleura sp13 | 12 | 0:07:01 |
| Tettigonia cantans | 37 | 0:58:10 | Stenobothrus lineatus | 19 | 0:34:27 | Chorthippus albomarginatus | 11 | 0:40:29 |
| Conocephalus fuscus | 36 | 0:53:34 | Clinopsalta autumna | 19 | 0:04:16 | Eupholidoptera schmidti | 11 | 0:09:40 |
| Chorthippus brunneus | 35 | 0:21:57 | Phaneroptera nana | 18 | 0:30:50 | Melanogryllus desertus | 11 | 0:25:24 |
| Decticus verrucivorus | 34 | 1:15:04 | Conocephalus dorsalis | 18 | 0:23:07 | Tylopsis lilifolia | 11 | 0:03:30 |
| Tettigonia viridissima | 33 | 0:27:26 | Platypleura sp10 | 17 | 0:17:55 | Ruspolia nitidula | 11 | 0:12:35 |
| Ephippiger diurnus | 31 | 0:39:51 | Chrysochraon dispar | 17 | 0:15:36 | Diceroprocta eugraphica | 11 | 0:05:07 |
| Nemobius sylvestris | 30 | 0:38:44 | Pholidoptera aptera | 16 | 0:10:55 | Platypleura sp12cfhirtipennis | 10 | 0:07:42 |
| Oecanthus pellucens | 29 | 0:30:32 | Eumodicogryllus bordigalensis | 16 | 0:10:56 | Omocestus petraeus | 10 | 0:09:22 |
| Gomphocerippus rufus | 28 | 0:29:38 | Platycleis albopunctata | 15 | 0:24:45 | Stauroderus scalaris | 10 | 0:20:43 |
| Pholidoptera griseoaptera | 27 | 0:14:07 | Atrapsalta corticina | 15 | 0:02:15 | Chorthippus vagans | 10 | 0:11:43 |
| Omocestus viridulus | 27 | 0:45:48 | Neotibicen pruinosus | 15 | 0:04:41 | Bicolorana bicolor | 10 | 0:09:19 |
| Gampsocleis glabra | 27 | 0:55:18 | Atrapsalta encaustica | 15 | 0:04:33 | Popplepsalta aeroides | 10 | 0:01:46 |
| Acheta domesticus | 24 | 0:56:48 | Metrioptera brachyptera | 14 | 0:20:56 | Atrapsalta collina | 10 | 0:01:20 |

## Data augmentation

Since the recordings varied in duration, they had to be divided into segments of a fixed length that could be fed into the network. A length of five seconds was chosen, as most calls were either short and rhythmical or long and static. Repeating sequences of longer than five seconds were not commonly observed in the dataset, therefore it was assumed that a length of five seconds would not eliminate species-specific rhythmic characteristics in the calls. Short files were looped until they reached five seconds in length. Longer files were sequentially spliced into chunks of five seconds, with an overlap of 3.75 seconds. When the splitting window reached the end of a file, the beginning of the recording was wrapped around to extend the chunk to five seconds, as long as the minimum remaining time of a chunk was at least 1.25 seconds.

For deep learning, it is standard practice to expand modest-sized training data through a synthetic process of audio augmentation, and we applied this to all three datasets. The training set of InsectSet32 was expanded with ten generations of audio augmentations using the python package "audiomentations" (github.com/iver56/audiomentations). The processing steps included "FrequencyMask", which erases a band of frequencies around a random center frequency, with bandwidth as a parameter that can be randomized within a defined range (0.06–0.22). This augmentation step was applied with a chance of 50%. After frequency masking, the signal was mixed with Gaussian noise, using the "AddGaussianSNR" function. The ratio of signal to noise was randomized between 25 and 80 dB. This ratio was tuned to range from barely noticeable addition of noise to heavy noise disturbance without obscuring the relevant audio information in noisy source recordings. This was applied to every file. After mixing with noise, the files were augmented with impulse responses (IRs) recorded in natural outside settings. This introduces reverberations and absorption characteristics of outside environments to the audio signal, simulating distance from the recording device. The IRs were selected from a dataset of recordings made in various locations at high sample rates [34]. Eleven IRs from three different outside locations (two forest locations, one campus location) were selected from this dataset and randomly applied during augmentation with a chance of 70%. The IR-processed files were mixed with their original version at random mix ratios to achieve additional variation in the severity of the effect, simulating varying distance from the recorder to the sound source.

For InsectSet47 and InsectSet66, online data augmentation was used, due to the vastly increased amount of audio material. From the package "torch_audiomentations" (github.com/asteroid-team/torch-audiomentations), the functions "AddColoredNoise" and "ApplyImpulseResponse" were used. Their parameters were tweaked to mimic the augmentations used in the smaller dataset. Unfortunately, a functionally similar function to the frequency masking used on the smaller dataset was not available in the package. As an alternative, the opportunity to vary the frequency distribution of the noise augmentation was used as an alternative to frequency masking. The frequency power decay was randomized between –2 and 1.5. The signal to noise ratio was randomized between 25 and 40 dB, with an overall probability of augmentation of 90%. Impulse responses were applied with a probability of 70%, with delay compensation enabled. The same IR files as in the smaller dataset were used [34] and mixed at a randomized mix ratio. Both augmentations were applied and randomized per example in a batch (Fig 2).

## Frontends

The frontends that were compared are the conventional mel spectrogram included in the python package torchaudio (MelSpectrogram) and the adaptive, waveform-based frontend LEAF [18]. The mel spectrograms were generated based on the audio waveforms before the

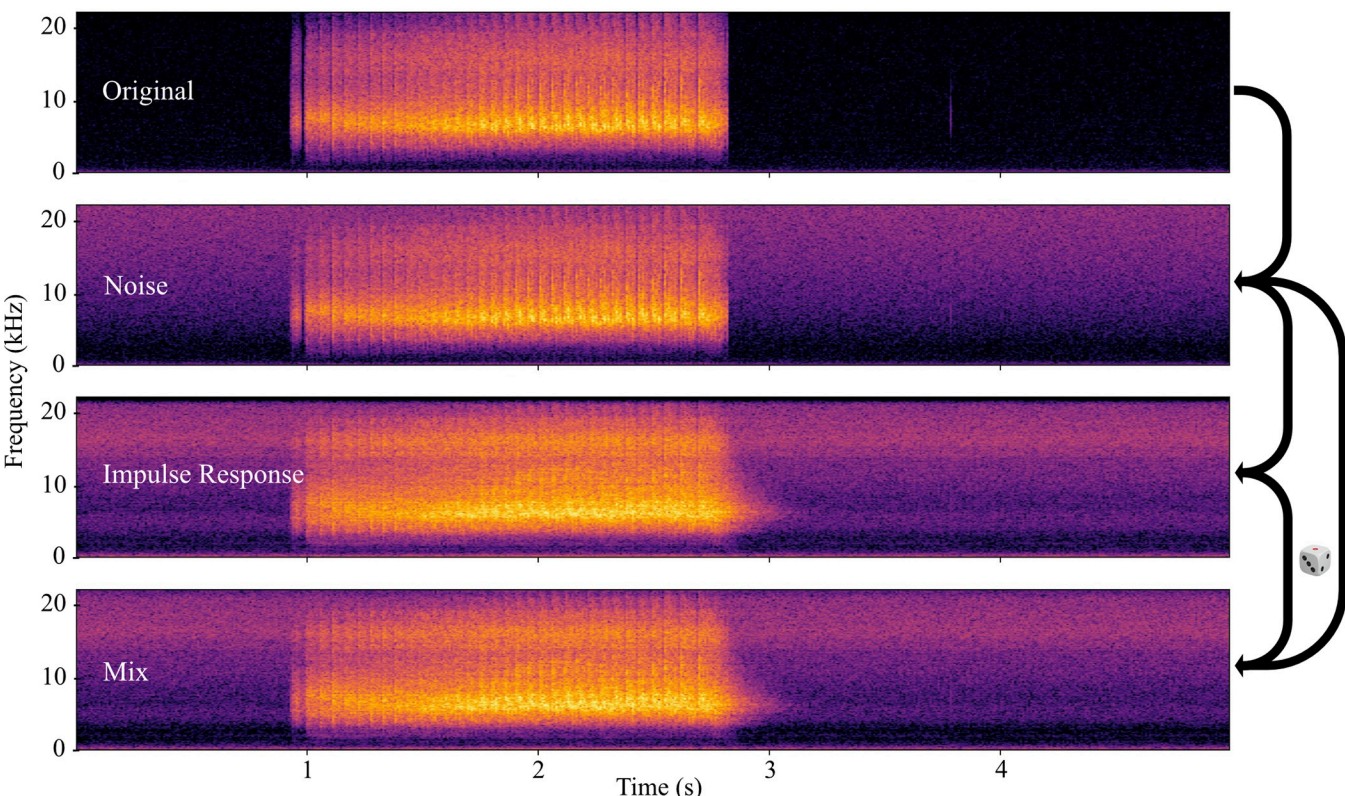

**Fig 2. Example of the data augmentation workflow used on the training set (InsectSet47 and InsectSet66). Noise is added at a randomized signal-to-noise ratio and frequency distribution.** Then an impulse response from an outdoor location is applied at a randomized mix ratio.

files were input into the convolutional network. When using the LEAF frontend, the full wave-forms were directly input to the network and then processed by the frontend, since many of its parameters like filter frequency and bandwidth, per-channel compression and normalization, and lowpass pooling can be learned and therefore need to be part of the network to benefit from gradient descent learning. The initialization parameters of the two frontends were defined as similarly as possible to create a fair comparison. The files were imported at a sample rate of 44.1 kHz. They were transformed from an input shape of [1; 220500] (one channel mono audio; 44.1 kHz for five seconds) to a representation shape of [1; 64; 1500] by the frontends, with 64 filter bands on the frequency axis and 1500 steps dividing the time axis. The window length was set at twice the length of the stride for both frontends (stride: 3.335 ms, window size: 6.67 ms). The filter bank used in the LEAF frontend was initialized on the same scale as the mel frontend, between 0 and 22.05 kHz. The inputs were combined into batches of 14 and fed into the network.

Additional tests were conducted to test the impact of the filterbank and PCEN components that make up the LEAF frontend. The models were trained on InsectSet47 and InsectSet66 using the same model architecture and LEAF frontend configuration as before, but the adjustment of either the filterbank or PCEN parameters during the training process were deactivated. This means that in the test case "leafFB" the filterbank parameters were adjusted during training, but the compression parameters of the PCEN component remained in the initialized state. In the test case "leafPCEN", the filterbank and temporal pooling parameters remained frozen in their initialized state, while only the PCEN compression parameters of the frontend were trained.

## Network

The network backend was adapted from a convolutional neural network created using pyTorch that was optimized for audio classification [28]. It consists of four convolutional layers (Conv2d) with rectified linear units (ReLU) and batch normalization (BatchNorm2d). After the convolutional layers, the feature maps were pooled (AdaptiveAvgPool2d) and flattened, and finally input into a linear layer (Linear) that returns a prediction value for each of the classes contained in the dataset. The highest prediction value was picked as the final predicted class for each training example. To avoid overfitting of the network on the small training dataset, dropout was implemented on the final linear layer (dropout rate of 0.4), as well as L2 regularization of the weights (weight decay of 0.001). The dropout rate was decreased to 0.23 for InsectSet47 and InsectSet66 since the models were underfitting as a result of the increased complexity of data. A fifth convolutional layer was added to the model for additional tests. Overall, the main model with four layers contains 28,319 trainable parameters that are adjusted during the training phase, with the inclusion of the LEAF frontend.

During the training process early stopping was employed, which evaluates the network performance after each epoch by running an inference step on the validation set. The loss value of the validation set was used to estimate how well the network would perform on the test set during final evaluation. Each time the validation loss decreased, the current network state was saved. If the validation loss did not decrease any further in eight consecutive epochs, the training was stopped and the final test evaluation was performed on the last saved network state from eight epochs earlier. The accuracy of the two approaches was determined by the percentage of correctly classified items in the test set, as well as the f1-score, precision and recall [12]. Due to the randomness included in the training process from dataset shuffling and network initialization, the training and evaluation outcomes can vary substantially between runs using the exact same parameters and datasets. To achieve a stable and comparable result on the small dataset, both models were computed five times each on InsectSet32 and three times each on InsectSet47 and InsectSet66. The best performing runs trained on InsectSet47 and InsectSet66 were trained again with an added fifth convolutional layer to test the effect of a larger model on the classification performance. All scripts used for preparing and classifying the data are publicly available on GitHub [35,36].

## Results

### InsectSet32

The median classification accuracy score for five runs using the mel frontend model was 62%, with scores for the different runs ranging between 57% and 67% (Table 4). The median classification accuracy for the LEAF models was 76% with a range from 59% to 78% (Table 4). The mel frontend achieved a median validation loss of 1.49, while the LEAF frontend had a lower median validation loss of 1.24 (Table 4). When looking at the additional performance metrics F1-score, recall and precision, even the worst performing LEAF run outperformed all of the mel runs (Table 4).

The majority of misclassifications (Fig 3) lied within the two biggest genera represented in InsectSet32, *Myopsalta* and *Platypleura* (5 and 14 species respectively, of 32 in total; Table 1). Species in these genera were most often misclassified as other members of their own genus. One particular species, *M. leona*, caused many misclassifications within its genus, despite being correctly classified itself. Similarly, within the genus *Platypleura*, the species labels *P. plumosa* and *P. sp12cfhirtipennis* were frequently assigned to other species of the same genus.

The confusion matrix showing the performance of the LEAF frontend reflects the overall better performance since it displays a clearer diagonal line of accurate classifications, with

**Table 4. Test and validation scores for all trained models with mel and LEAF frontends on insect sound datasets of three different sizes.** The median as well as the lower and upper limits are reported from training multiple runs of the same model with different randomization seeds and four convolutional layers (five runs each for InsectSet32, three runs each for InsectSet47 and InsectSet66). The best performing models were also trained with an additional convolutional layer, indicated by the number in the model name.

| Dataset | Model | Test | | | | Validation | |
|---|---|---|---|---|---|---|---|
| | | Accuracy | F1-score | Recall | Precision | Accuracy | Loss |
| InsectSet32 | mel-4 | 0.62 0.57–0.67 | 0.52 0.47–0.56 | 0.53 0.49–0.58 | 0.61 0.52–0.64 | 0.60 0.57–0.65 | 1.49 1.37–1.68 |
| | LEAF-4 | 0.76 0.59–0.78 | 0.66 0.61–0.69 | 0.68 0.60–0.71 | 0.70 0.67–0.73 | 0.71 0.61–0.76 | 1.24 1.00–1.40 |
| InsectSet47 | mel-4 | 0.77 0.70–0.77 | 0.66 0.56–0.67 | 0.66 0.57–0.67 | 0.69 0.63–0.74 | 0.75 0.71–0.77 | 0.98 0.92–1.14 |
| | LEAF-4 | 0.81 0.79–0.83 | 0.71 0.71–0.77 | 0.72 0.71–0.76 | 0.77 0.74–0.83 | 0.84 0.83–0.86 | 0.72 0.72–0.74 |
| | mel-5 | 0.85 | 0.78 | 0.79 | 0.81 | 0.83 | 0.69 |
| | LEAF-5 | 0.86 | 0.81 | 0.81 | 0.85 | 0.88 | 0.58 |
| InsectSet66 | mel-4 | 0.78 0.75–0.78 | 0.66 0.65–0.69 | 0.66 0.64–0.69 | 0.73 0.73–0.74 | 0.76 0.76–0.76 | 0.98 0.97–0.98 |
| | LEAF-4 | 0.80 0.79–0.81 | 0.68 0.67–0.71 | 0.68 0.67–0.70 | 0.77 0.74–0.77 | 0.83 0.80–0.84 | 0.81 0.79–0.86 |
| | mel-5 | 0.82 | 0.74 | 0.74 | 0.80 | 0.81 | 0.82 |
| | LEAF-5 | 0.83 | 0.76 | 0.77 | 0.81 | 0.85 | 0.73 |

fewer incorrect classifications around it (Fig 4). Both test files of the species *Brevisiana brevis* were incorrectly classified as *Platypleura haglundi*. The species *P. intercapedinis* (two test files) and *P. sp11 cfhirtipennis* (one file) were never correctly classified either but confused with different species of the same genus. The concentration of misclassifications in the two largest genera *Myopsalta* and *Platypleura* is much less pronounced compared to the mel frontend run. In particular, the performance within *Myopsalta* is substantially better (Figs 3 and 4).

The filters employed by the LEAF frontend were initialized on a scale closely matched to the mel scale but were adjusted in center frequency and bandwidth during training on Insect-Set32 (Fig 5). After sorting the filters by their center frequencies, they continue to largely adhere to the initialization curve (Fig 5C and 5F). Without sorting however, it is clear that many filters were adjusted from their original position (Fig 5B and 5E). Substantial changes in the frequencies of several filters occurred around 2 kHz and above 15 kHz, where some filters were adjusted by up to several kilohertz, especially with the highest filter at initialization being shifted from 22.05 kHz down to approximately 13 kHz (Fig 5B). The ordering along the frequency axis is heavily disturbed, since the center frequencies do not steadily increase with increasing filter number, as was the case on the initialized scale (Fig 5B and 5E). This means that in the LEAF output matrices, adjacent values on the axis containing frequency information do not necessarily represent adjacent frequency bins, which is usually the case when using hand-crafted representations such as mel filter banks. Filter density increased around 0.85 kHz (see Fig 5D, ≈ 900 mel) and between roughly 14–15 kHz (Fig 5B), but slightly decreased between 18 and 20 kHz (Fig 5B) and around 2.4 kHz (see Fig 5D, ≈ 1700 mel). Four filters are located close to zero mel/kHz after training, leaving a gap up to approximately 500 mel (≈ 0.4 kHz), where the very lowest insect sound frequencies occur in this dataset (Fig 5D).

## InsectSet47

On the expanded InsectSet47, the median classification performance achieved with the mel frontend was 77% and a median loss of 0.98 on the validation set. This is a substantial

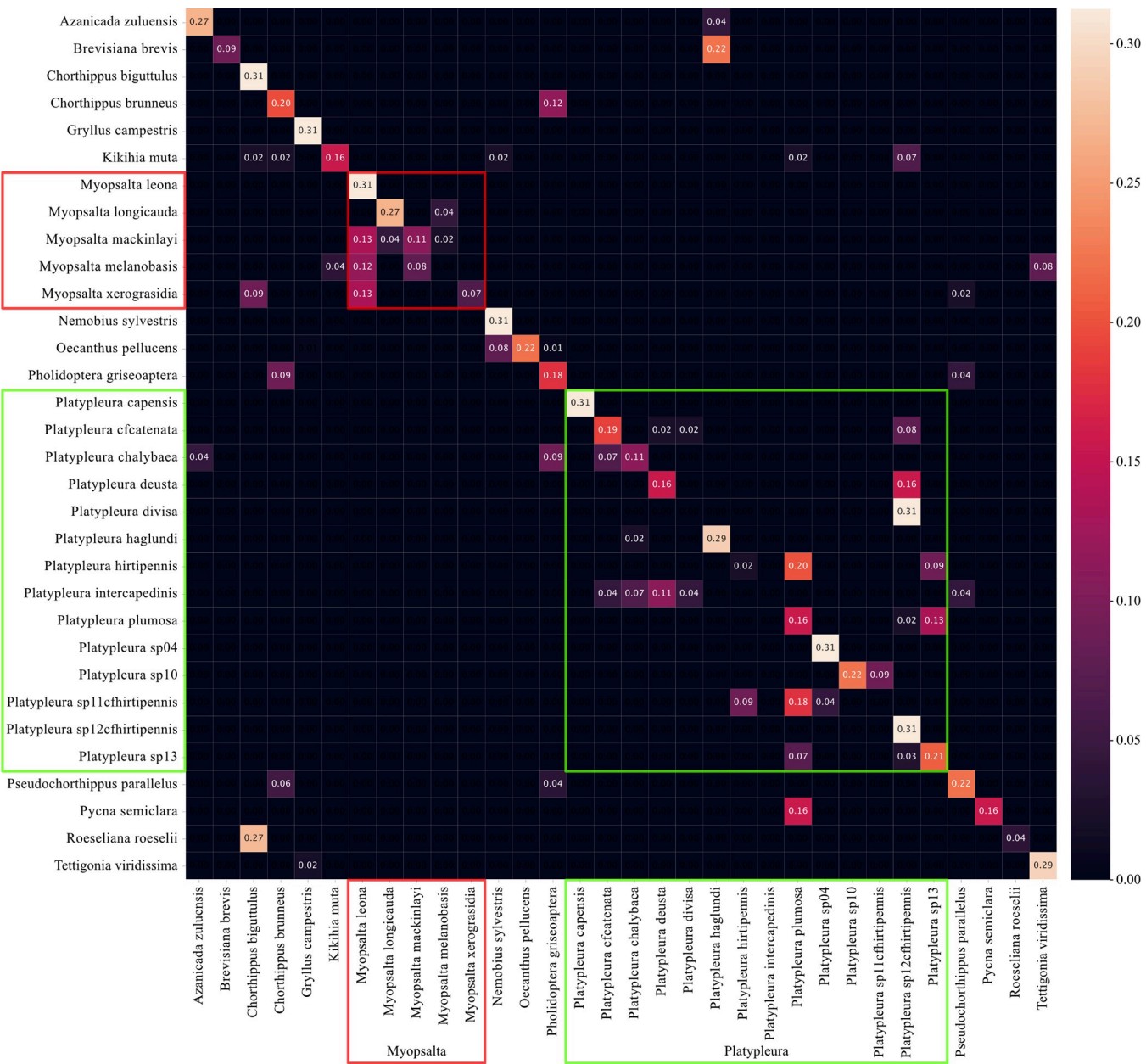

**Fig 3. Classification outcome for all 32 species in the test set using the best run of the mel frontend performing at 67% classification accuracy.** The vertical axis displays the true labels of the files, the horizontal axis shows the predicted labels, sorted alphabetically. Classifications within the two biggest genera *Platypleura* (green) and *Myopsalta* (red) are highlighted for comparison to the LEAF confusion matrix.

improvement in performance compared to InsectSet32, despite the increased number of species (Table 2). The LEAF frontend gained a less substantial increase in classification performance, but still outperformed the mel frontend in all three runs with a median 81% classification accuracy and substantially lower loss of 0.72 (Table 4). The difference between the frontends was smaller overall however, compared to InsectSet32. The models trained with an additional convolutional layer improved even further in performance. The mel frontend gained a larger increase in classification performance from this, reaching 85%, while LEAF performed only slightly better at 86% (Table 4).

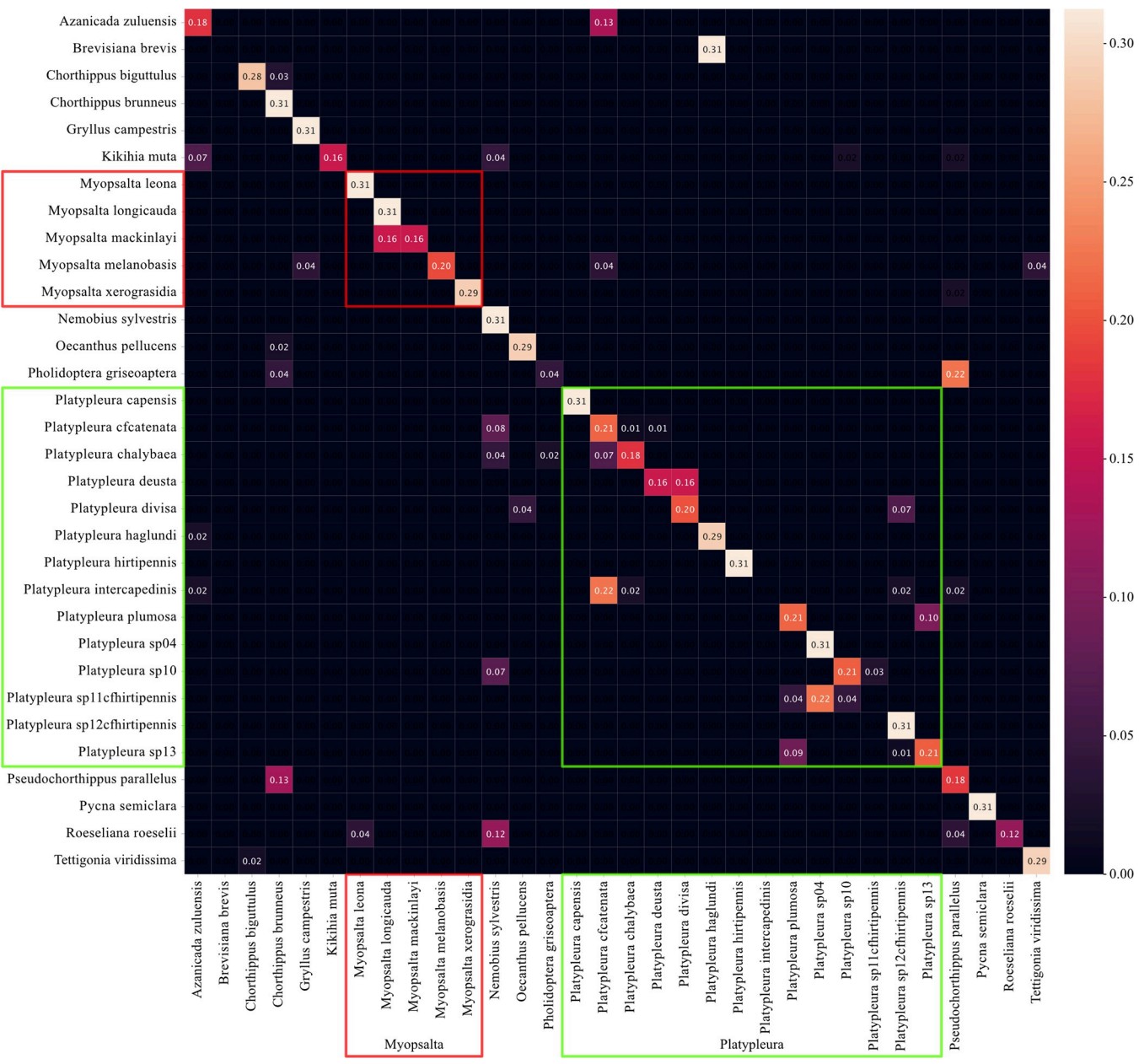

**Fig 4. Classification outcome for all 32 species in the test set using the best run of the LEAF frontend performing at 78% classification accuracy.** The vertical axis displays the true labels of the files, the horizontal axis shows the predicted labels, sorted alphabetically. Classifications within the two biggest genera *Platypleura* (green) and *Myopsalta* (red) are highlighted for comparison to the mel confusion matrix.

Using both frontends, misclassifications between the groups of Orthoptera and Cicadidae were negligible (S1 and S2 Figs). In general, classification errors appeared more frequently with closely related species. The LEAF frontend was able to improve performance over the mel frontend by reducing the large number of misclassifications in the genus Acrididae (S1 and S2 Figs). In the genus Playtpleura, nearly all audio examples of two species (*P. sp12cfhirtipennis* and *P. sp13*) were classified as *P. plumosa* by the mel frontend (S1 Fig). The LEAF frontend managed to reduce the incorrect classifications to *P. plumosa* roughly by half, by compromising half of the correct classifications of that species (S2 Fig).

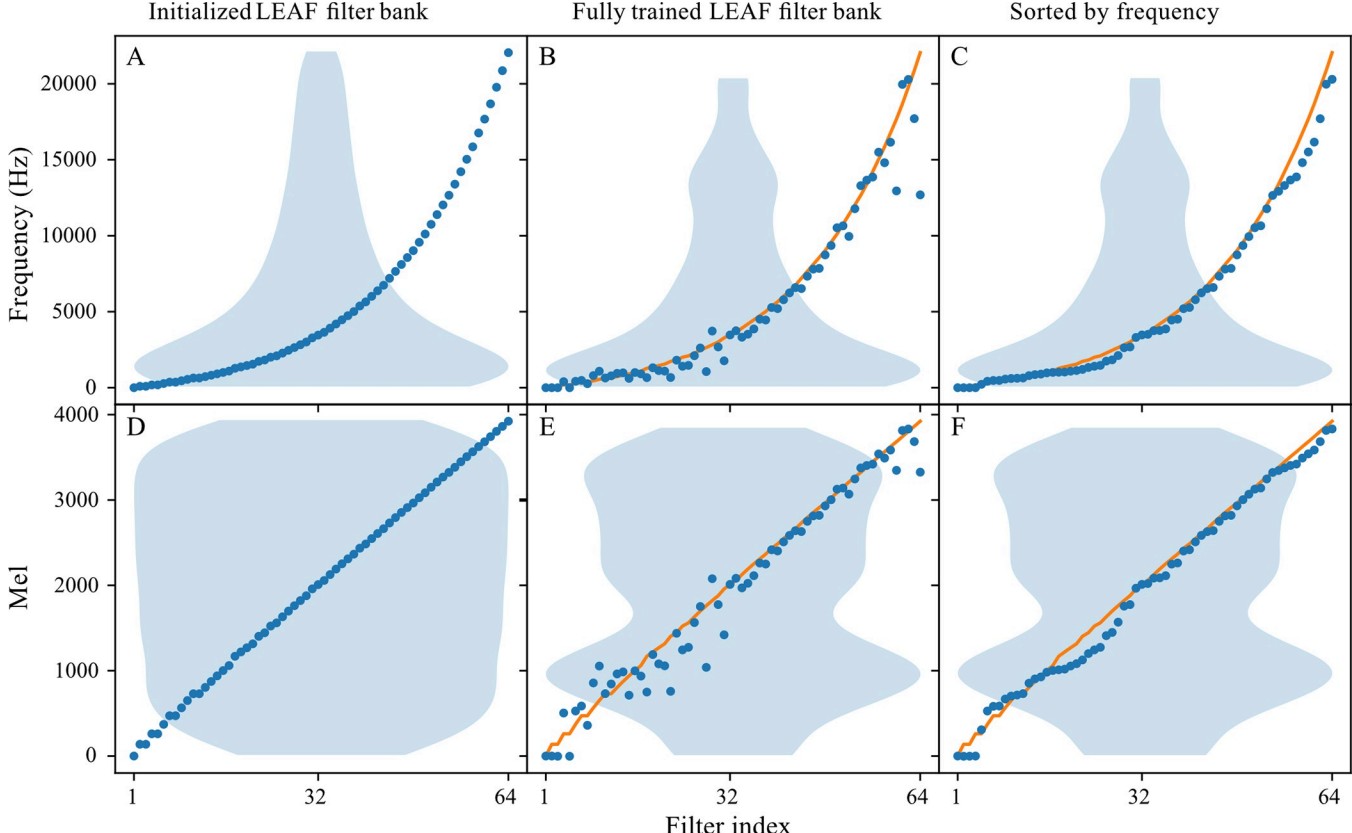

**Fig 5. Center frequencies of all 64 filters used in the best performing LEAF run on InsectSet32.** Plots A and D show the initialization curve before training, which is based on the mel scale. Plots B and E show the deviation of each filter from their initialized position after training. Plots C and F show the filters sorted by center frequency, and demonstrate the overall coverage of the frequency range, but do not represent the real ordering in the LEAF representations. Violin plots show the density of filters over the frequency spectrum, the orange line shows the initialization curve for comparison.

### InsectSet66

The models trained on InsectSet66 showed similar results to InsectSet47, again despite the increase in the number of classes. The mel frontend slightly improved its median classification performance from 77% to 78% on this larger dataset, while the LEAF performance decreased from 81% to 80% (Table 4). The median loss stayed on the same level as on InsectSet47 for the mel frontend with 0.98, but increased for the LEAF frontend from 0.72 on InsectSet47 to 0.81 on InsectSet66 (Table 4). The performance, when trained with five convolutional layers, improved again for both frontends, where the LEAF frontend only had a small advantage with 83% compared to the 82% reached with the mel frontend (Table 4). For both frontends, incorrect classifications of Orthoptera species as Hemiptera are almost non-existent. Classifications of Hermiptera as Orthoptera do appear, but are rare (S3 and S4 Figs). In general, misclassifications appear most often within the genera. The confusion matrices of LEAF and mel do not show obvious differences or trends, likely since the overall classification performance is similar.

### leafPCEN

The training of the leafPCEN frontend, which retains the trainable PCEN part of LEAF, but freezes its filterbank and pooling parameters, did not succeed. The validation accuracy and

**Table 5. Test and validation scores for the trained models using the leafFB frontend.** The median as well as the lower and upper limits are reported from training three runs of the same model with different randomization seeds and four convolutional layers.

| Dataset | Model | Test | | | | Validation | |
|---|---|---|---|---|---|---|---|
| | | Accuracy | F1-score | Recall | Precision | Accuracy | Loss |
| InsectSet47 | leafFB-4 | 0.81 0.72–0.83 | 0.73 0.60–0.75 | 0.73 0.60–0.75 | 0.79 0.74–0.82 | 0.84 0.73–0.86 | 0.74 0.71–1.14 |
| InsectSet66 | leafFB-4 | 0.79 0.70–0.81 | 0.67 0.59–0.69 | 0.67 0.59–0.68 | 0.72 0.69–0.76 | 0.82 0.72–0.84 | 0.79 0.79–1.22 |

loss values showed large spikes and did not converge effectively. Three runs were trained on InsectSet47, but a median classification accuracy on the test set of only 71% was reached, which is substantially worse than the standard LEAF or even mel frontends performances (Table 4). Because of this, the frontend was not trained on InsectSet66.

### leafFB

The leafFB frontend, which employed a trainable filterbank, but used the initialized PCEN component of the LEAF frontend, performed better than the leafPCEN frontend, and converged despite occasional spikes of the accuracy and loss values during training. On Insect-Set47, leafFB reached a median classification accuracy of 81% and a median loss value of 0.74 (Table 5), performing slightly better than the standard LEAF frontend (Table 4). On Insect-Set66, the performance decreased to a median of 79% classification accuracy and a median loss of 0.79 (Table 5), which is slightly worse than the LEAF frontend (Table 4). On both datasets, more variation in performance between the runs was observed, meaning that some leafFB runs did perform substantially worse than LEAF (Tables 4 and 5).

### Discussion

The focus of this work was mostly to compare a traditional handcrafted feature extraction method (mel) against an adaptive and waveform-based method (LEAF), while also testing the viability of deep learning methods to classify insect sounds, specifically of Orthoptera and Cicadidae. Three datasets were used for this comparison, with increasing number of audio files, as well as numbers of species. In all settings, the adaptive frontend LEAF outperformed the mel frontend (Table 4), by adjusting its filter bank and compression parameters to fit the data (Fig 5). This effect was most pronounced on the smallest dataset InsectSet32, where LEAF reached a classification accuracy of 78%, compared to 67% using mel (Table 4). On the expanded dataset InsectSet47, the performance of both frontends improved in comparison to InsectSet32, despite the increased number of species. This is likely due to the much higher number and length of audio examples, allowing the models to generalize better on unseen data. The difference in performance between the frontends decreased however. The performance of the mel frontend on largest dataset InsectSet66 overall remained roughly on the same level as on InsectSet47, even though a substantial number of species was added, but not a large amount of audio material (Table 3).

Since the performance seemed to plateau at this level, we hypothesized that the complexity of the backend classifier was reaching a limit and was not able to process the full amount of information contained in the larger datasets. This could have obscured an advantage in the feature extraction performance by the frontends. To rule this out, more tests were conducted on InsectSet47 and InsectSet66 by adding an additional convolutional layer to the models, with the expectation that this would allow the LEAF performance to increase more than the mel performance. This modification led to increased classification performance in all cases, but

actually decreased the difference between the frontends (Table 4). On InsectSet47, the mel frontend improved substantially from 77% to 85%, while the LEAF frontend only improved from 83% to 86% (Table 4). On InsectSet66, the mel frontend improved from 78% to 82% and LEAF from 81% to 83% (Table 4). This could mean that the ability of the LEAF frontend to adjust feature extraction parameters might be more relevant when there is only a limited number of audio examples. But the characteristics of the audio data could also affect how LEAF performs in comparison to mel.

In similar comparisons on more human-centric audio classification tasks (language, emotion, birdsong, music etc.), LEAF outperformed mel spectrograms on a diverse range of tasks, but not all, and in many cases by smaller margins than in this comparison [18]. Since the sounds in this application are very different in structure and frequency content from human-associated sounds, the difference in performance between LEAF and mel was expected to be larger than in the previous comparisons. LEAF can learn a large number of parameters and adapt to the input data, while the mel frontends parameters are completely fixed and not necessarily ideal when not used with human sounds. The relevant information in insect sound is largely located in the higher frequency spectrum (above 5 kHz), where mel spectrograms are more imprecise due to increasingly wider pooling of frequencies. The LEAF frontend adjusted filter center frequencies and bandwidths, as well as compression and time-pooling parameters to better fit the data and reveal details that could be obscured by the mel frontend fixed parameters (Fig 5).

The confusion matrices generated from InsectSet32 shed some light on where the differences in performance lie between the two approaches (Figs 3 and 4). Using the mel frontend, the majority of incorrect classifications was found between species of the genus *Platypleura*, which represents almost half of the species included in the dataset with 14 out of 32, and in the second largest genus *Myopsalta*, with five species (Table 1). These two groups make up the majority of the species in InsectSet32 and it is therefore more likely for them to contain a majority of the misclassifications. However, the fact that many of their false classifications were within species of the same genus suggests that their sounds could be similar in structure and hard for the network to distinguish. The trained parameters of the LEAF frontend led to much better performance in these two genera than the mel frontend, since there were fewer false predictions within these genera while false predictions outside of these genera remained roughly the same (Figs 3 and 4). The confusion matrices generated from InsectSet47 and InsectSet66 did not reveal clear differences between mel and LEAF, since the overall performance of the frontends was much more similar compared to InsectSet32 (Figs 1–4). It is possible that due to the larger diversity of species and genera, the LEAF frontend did not fine-tune its parameters to distinguish between specific sound characteristics of closely related species to the same extent as observed in InsectSet32. In this dataset, especially the species in the largest genus *Platypleura* produce very similar sounds. They are generally noisy with most spectral energy between 7 and 10 kHz and contain subtle frequency modulations with rates of roughly 20 to 50 Hz. Overall, the *Platypleura* songs in InsectSet32 are very static in frequency content and volume, and are not easily distinguishable. Additionally, some of the labels included in this dataset are from populations that have not fully been determined to the species level, which could mean that some of them represent subpopulations of the same species or very closely related, undescribed species. The adaptive nature of the LEAF frontend with its multiple tunable parameters might have been especially advantageous for this task. The mel frontend might have been able to form strong enough representations for the more diverse range of species included in InsectSet47 & 66. This would explain the decreased advantage LEAF had in combination with the larger model (Table 4): Increased detail in the audio representations was not specifically needed, but more complexity in the model as a whole, which LEAF also

provides. Repeating these experiments with datasets that contain groups of insects that produce very similar sounds and datasets with a diverse selection of species could shed some more light on these findings.

The overall coverage of filters over the frequency spectrum was not substantially changed during training of the LEAF frontends. When looking at the filter distribution after training, the filters still mostly lie close to the initialization curve that was based on the mel scale (Fig 5C and 5F). While changes in filter density occurred in some frequency bands, a dramatic shift of all filters shifting to higher frequencies or a change to a completely different curve was not observed. When considering the changes of every individual filter however, it is clear that many filters changed position quite substantially, sometimes by several thousand Hertz (Fig 5B and 5E). The ascending order of filter bands along the frequency axis is heavily disturbed after training, meaning that adjacent rows in the LEAF output matrices do not necessarily contain adjacent bands in the frequency domain. Interestingly, this was not observed in the original paper introducing the LEAF frontend [18] nor in a paper improving the performance of the frontend [37]. In these studies, LEAF frontends were trained on the AudioSet [38] and SpeechCommands datasets [39] at sample rates of 16 kHz. The resulting filter bank configurations still closely followed the initialization curve after training and the ordering along the frequency axis was conserved [18,37]. This was interpreted as a demonstration that the mel scale is a strong initialization curve for these tasks, with the learnable filter parameters in the LEAF frontend mostly providing an opportunity for adapting to a slightly more appropriate frequency range [18,37].

The AudioSet dataset contains many human-centric sounds such as speech and music, as well as a diverse set of environmental sounds, animal sounds and more, with 527 classes and multiple labels per recording [38]. The SpeechCommands dataset contains over 100,000 samples of spoken words [39]. Perhaps such a diversity of sounds and classes, as well as the use of a much lower sample rate of 16 kHz [18] constrained the adjustment of filter frequencies compared to the much smaller datasets used in our comparison which focused on a more fine-grained classification task. It is also possible that ordering along the frequency axis is more important for classifying sounds that contain defined harmonic structures such as human speech, music, instruments or birdsong. The often noisy and inharmonic sounds produced by Orthoptera and Cicadidae might not require this due to their more uniform and comparably undefined sonic structure over the spectrum.

Since the LEAF frontend is a combination of a learnable filter bank and learnable PCEN compression, we wanted to determine the influence of the individual components on the improved performance over the mel frontend. Especially since the overall filter bank curve was not adjusted as strongly as expected and because PCEN as a replacement for the conventional log-compression has been shown to be advantageous in some, but not all cases for classifying environmental sounds [40–42]. A modification of the LEAF frontend with disabled training of the filterbank and temporal pooling parameters, but trainable PCEN parameters was tested, called leafPCEN. This frontend should essentially function like a standard mel frontend with an added trainable PCEN component since the initialized LEAF filterbank functions like a mel filterbank. Surprisingly, leafPCEN did not train successfully and even performed worse than the normal mel frontend (Table 5). It has been observed in previous work that in some applications, depending on the signal and background noise characteristics, trainable PCEN parameters can fail to converge on ideal values and lead to suboptimal feature extraction [40,42]. It appears that in the LEAF frontend, without the trainable filterbank, the PCEN component can be unstable and collapse into poor configurations. The leafFB frontend, which retains the trainable filterbank and pooling of LEAF, but disables training on the PCEN compression parameters, performed at roughly the same level as the standard LEAF frontend, although

with more variation between the runs (Tables 4 and 5). This suggests that the adjustment of the filterbank parameters specifically lead to a better configuration than the standard mel frontend and increased the classification performance.

The high occurrence of adjustments and shuffling of individual LEAF filters could justify testing different initialization curves than the mel scale. While this scale has been shown to be robust and advantageous for classifying human-centric sounds [18], it might not be the ideal initialization curve for insect sounds. The theoretical justifications for the use of the mel-scale do not apply to the much higher frequency ranges and faster temporal patterns of insect sounds. Perhaps the filter distributions learned in this study are local optima that could be reached from the mel curve as a starting point, but expert-designed initialization curves could allow the frontend to reach a better and more generalizable filter distribution for insect sounds in a shorter amount of training time, which would be advantageous. One experiment testing a different initialization curve was conducted with randomized center frequency values that were sorted in ascending order [37]. During training, the filter values were adjusted to a more appropriate frequency range for the data, but the overall performance was lower than when using a mel initialization curve, when tested on the SpeechCommands dataset [37,39]. This, again, shows that the mel scale is very robust and useful for human sounds, but also that LEAF can learn useful filter distributions even when not initialized on an ideal scale [37]. This further justifies the exploration of alternative initialization scales for usage of the LEAF frontend with non-human sounds.

To achieve further improvement of classification performance, especially if machine learning methods are going to be implemented in species conservation efforts, larger and more diverse datasets should be the focus. In this work, up to 66 species were represented, with a minimum of 10 recordings per class. This could be a realistic number of species for monitoring specific environments or even larger geographic areas. But for future implementations, existing datasets are not sufficient and have to represent all species that occur in the environments where automatic classification methods are going to be deployed. The number and length of recordings per species should also be increased to achieve better representations of the natural variations in the insects sounds. If datasets with higher sample rates are going to be used for classification, conventional mel spectrogram frontends may prove to be even less useful compared to adaptive frontends. Especially for species that produce sounds entirely within the ultrasonic range, which are common in Orthoptera and some Cicadidae [43], the lower resolution in high-frequency bands would be increasingly disadvantageous compared to adaptive frontends.

While compiling the datasets for this work, special attention was paid to exclude recordings with low audio quality and especially recordings that contain sounds from multiple insect species, even if other species were barely noticeable in the background. Since many of the recordings from the source databases are submissions from citizen-scientists that did not meet the quality standards for this work, a large amount of audio material was not included in these datasets. Lowering the quality standards would allow the inclusion of many more species and audio examples. Whether this would be beneficial remains to be tested, since the added amount of audio material could offset the negative effects of lower quality recordings.

Considering the relatively simple network architecture and small datasets, these results are encouraging for future applications with high potential for further improvements through optimizing model parameters and diversifying datasets. The advantage in performance by using LEAF, despite being small in some cases, identifies adaptive frontends as a potentially valuable replacement for approaches with hand-crafted parameters to extract features for insect audio classification. Before these methods can be applied in conservation efforts, datasets need to be increased in size and species diversity, and the networks that are used must be improved to reach higher overall accuracy. These methods also need to be integrated with sound-event detection

methods to automatically identify relevant clips from longer automatic recordings. This work presents a first step for optimizing an important part of the classification network and shows encouraging results and methods for successful future implementations of this technology.

## Supporting information

**S1 Fig. Classification outcome for all 47 species in the test set using the best run of the mel frontend performing at 77% classification accuracy.** The vertical axis displays the true labels of the files, the horizontal axis shows the predicted labels, grouped into order, family and genus.
(TIFF)

**S2 Fig. Classification outcome for all 47 species in the test set using the best run of the LEAF frontend performing at 83% classification accuracy.** The vertical axis displays the true labels of the files, the horizontal axis shows the predicted labels, grouped into order, family and genus.
(TIFF)

**S3 Fig. Classification outcome for all 66 species in the test set using the best run of the mel frontend performing at 78% classification accuracy.** The vertical axis displays the true labels of the files, the horizontal axis shows the predicted labels, grouped into order, family and genus.
(TIFF)

**S4 Fig. Classification outcome for all 66 species in the test set using the best run of the LEAF frontend performing at 81% classification accuracy.** The vertical axis displays the true labels of the files, the horizontal axis shows the predicted labels, grouped into order, family and genus.
(TIFF)

**S5 Fig. Center frequencies of all 64 filters used in the best performing LEAF run on Insect-Set47.** Plots A and D show the initialization curve before training, which is based on the mel scale. Plots B and E show the deviation of each filter from their initialized position after training. Plots C and F show the filters sorted by center frequency, and demonstrate the overall coverage of the frequency range, but do not represent the real ordering in the LEAF representations. Violin plots show the density of filters over the frequency spectrum, the orange line shows the initialization curve for comparison.
(TIFF)

**S6 Fig. Center frequencies of all 64 filters used in the best performing LEAF run on Insect-Set66.** Plots A and D show the initialization curve before training, which is based on the mel scale. Plots B and E show the deviation of each filter from their initialized position after training. Plots C and F show the filters sorted by center frequency, and demonstrate the overall coverage of the frequency range, but do not represent the real ordering in the LEAF representations. Violin plots show the density of filters over the frequency spectrum, the orange line shows the initialization curve for comparison.
(TIFF)

## Acknowledgments

We kindly thank Baudewijn Odé and Ed Baker for the use of their sound collections, as well as the contributors to Xeno Canto and iNaturalist.

## Author Contributions

**Conceptualization:** Marius Faiß, Dan Stowell.

**Data curation:** Marius Faiß.

**Formal analysis:** Marius Faiß.

**Funding acquisition:** Dan Stowell.

**Investigation:** Marius Faiß.

**Methodology:** Marius Faiß, Dan Stowell.

**Resources:** Dan Stowell.

**Software:** Marius Faiß.

**Supervision:** Dan Stowell.

**Validation:** Marius Faiß.

**Visualization:** Marius Faiß.

**Writing – original draft:** Marius Faiß.

**Writing – review & editing:** Marius Faiß, Dan Stowell.

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
