## [Decision Letter · Decision Letter 0]

17 Jul 2023

Dear Dr Stowell,

Thank you very much for submitting your manuscript "Adaptive representations of sound for automatic insect recognition" for consideration at PLOS Computational Biology. As with all papers reviewed by the journal, your manuscript was reviewed by members of the editorial board and by several independent reviewers. The reviewers appreciated the attention to an important topic. Based on the reviews, we are likely to accept this manuscript for publication, providing that you modify the manuscript according to the review recommendations.

Sincerely,

Ricardo Martinez-Garcia

Academic Editor

PLOS Computational Biology

James O'Dwyer

Section Editor

PLOS Computational Biology

Reviewer's Responses to Questions

**Comments to the Authors:**

Reviewer #1: Very nice paper. Two weaknesses

1) The authors seem to assume that deep learning is the answer. However, it is not clear why that should be true. In the somewhat related problem of classifying mosquitoes by flight sounds, many folk have tried deep learning, but simple nearest neighbor on the right features or a bayes classifier still seems state of the art.

2) The list of many datasets gives the paper a bit of a laundry list feeling.

“Many insect species produce characteristic sounds that can easily be detected and recorded without large cost or effort.”, Many species, but a very small subset of the number of insect species.

“Using deep learning methods, insect sounds from field recordings could be automatically detected and classified to monitor biodiversity and species distribution ranges.”

True, but you could do it without deep learning too, right? Why the instant jump to deep learning?

There have been some other research efforts on machine learning of insect sounds using spectrograms [a]. I am not saying you have to compare to these, but it would be at least a good idea to do a more detailed literature search and put your work in context.

“For example, ambient temperature during the recording can influence the frequency..” If you where to classify using a Bayesian classifier, you could condition the classifier on temperature. See the mosquito classification papers of Batista or Keogh etc.

In fig 1, it is odd the x-axis limits are 0 to 0.4901. Why not a round number, why that four digit precision?

“compiled from private collections of 186 Orthoptera and Cicadidae recordings (Orthoptera dataset by Baudewijn Odé and Cicadidae 187 dataset by Ed Baker, both unpublished)”

You need to make these available to the reviewers.

Fig 3 would be so much clearer if the cells with “0.00” were left empty of text.

[a] Yuan Hao, Bilson Campana, Eamonn Keogh. Monitoring and Mining Insect Sounds in Visual Space. SDM 2012

Reviewer #2: This was a very well written article. The authors provide a lot of detail making the research accessible to potential readers who might have a strong background within the topic discussed. The flow of ideas was good and the manuscript was easy to read.

Here are a few minor suggestions, below. It would be great if the authors could address these minor points which should not take long.

Line 206: could the authors please make it clear if the training, validation and testing files are completely independent, and that two or more spectrograms from a single audio file cannot be found in training/testing. The authors make this statement on the next dataset, but not for InsectSect32.

Line 305-311, the authors provide a bit less implementational details related to how they used IRs, compared to the amount of detail provided in general. It would be great if a bit more implementational detail was provided here, so that this matches with the rest of the manuscript.

Line 360, reference [22] is not the most academic :) Perhaps the authors could find a peer reviewed study that used a simple CNN?

Figure 3 (and all the other confusion matrices): please excuse me, but I have some trouble interpreting this plot. Typically, a confusion matrix is normalised (values between 0 and 1), or the raw values are provided. I can't seem to add up the values to one (1) in terms of species X being misclassified as Y, Z , ... along the rows. I'm sure there is a simple explanation, could the authors please clarify.

Reviewer #3: This manuscript presents the results of using a new adaptive frontend as input to convolutional neural network classifiers and compares these results to a more traditional mel-spectrogram frontend. The methods are well explained and the results illustrate the potential of this new method. There are a few places where more detail is needed (see my detailed comments). There are many inconsistencies in the use of tenses throughout the manuscript. I have pointed many of these out, but I recommend a very careful reading of the manuscript with this in mind. Once these things are addressed, I recommend publishing this interesting paper.

Abstract

The abstract provides a good summary of what was done but would benefit from including some quantitative results. For example, how many species were included in the classifiers? How much better did LEAF do than the standard representation of sound?

Lines 49-51: This sentence makes it sound like stridulation is the only way that sounds are produced by insects. Please modify so that it is clear that this is just one way that insects produce sound.

Line 61: What are the conventional monitoring methods used for insects? It would be helpful to mention that here.

Line 66: The phrase ‘since they can function as indicator species’ doesn’t really make sense at the end of this sentence. This seems to be a separate idea related to why it is important to be able to monitor them effectively, not to why they can be monitored acoustically.

Line 117: There does not need to be a comma after the word ‘parameters’

Figure 1: Spectrogram B needs labels on the x-axis

The introduction provides useful background and explains things well. It would be helpful to also include a brief description of the typical characteristics of sounds produced by Orthoptera and Cicadidae species that are included in the classifiers.

Lines 179-180: Please provide more detail about the neural network. Since this and the frontends are central to your results, more information than just a citation needs to be included.

Lines 190-191: How did you know what species were in the recordings? Please provide some information about how the recordings were made. What kind of equipment was used to make recordings? Where were the recordings made? Were they collected from wild insects or were they made in a laboratory setting?

Line 196: Is 40 seconds or 1 minute of data enough to characterize the sounds produced by a species? Please provide some explanation of why you feel confident that this is enough data. I question whether it is a large enough dataset.

Lines 213 and 249: I agree that citizen science recordings are a great resource, but when developing classifiers, it is crucial to use recordings from known species. Did you do any quality checking to ensure the accuracy of the species labels for the recordings that you used? If so, please describe this process. If not, please explain why this was not necessary.

Line 231: recommend changing ‘less species’ to ‘fewer species’

Lines 274 – 279: This should all be written in the past tense.

Line 284: Please change ‘can be fed’ to ‘could be fed’

Line 285: Please define ‘short and rhythmical’ and ‘long and static’. How short is short? How long is long?

Line 287-288: would looping short files until they reach 5 seconds not result in over-representing certain sound-types or noise environments?

Line 314: why did you not also use torch_audiomentations for the smaller dataset?

Lines 370-372: this sentence should be written in the past tense. The paragraph that follows should also be written in the past tense

Please make re-write the results section so that it is all in the past tense.

Figure 3 and Figure 4: Please provide a colour scale bar or describe in the figure caption what the different colours in the confusion matrices represent. Also, I assume that the green boxes denote Playtypleura and Myopsalta, but this should be stated in the figure. It would be helpful to have different coloured boxes for each genus for those readers who aren’t familiar with which is which.

Line 428: Please change ‘less incorrect classifications’ to ‘fewer incorrect classifications’

Lines 428-431: it would be helpful to state sample sizes for the species that you mention here.

Line 433: I recommend changing ‘run, especially’ to ‘run. In particular,’. Also, when you say ‘significantly better’ – is this statistically significant? If so, please provide the test and p-value. If not, I suggest rewording slightly and using a word other than significantly.

Line 474: Same comment as above regarding the word ‘significant’

Line 478: I suggest changing ‘decreased overall’ to ‘was smaller overall’

Line 479: ‘layers’ should be singular

Line 503: It’s not clear what you mean by ‘classifications in the opposite direction’

Line 520: I recommend removing the word ‘managed to’

Table 5: Is there a digit missing in the lower limit of accuracy for the InsectSet66 dataset?

Lines 560-562: could you go into a little more detail about this point? It is an important finding and I feel that it should be discussed in a bit more detail.

Lines 579-582: This sentence and much of this paragraph should be written in the past tense.

Lines 585-586: This seems like a very likely explanation. Have you looked at the sounds produced by some of these species to see if this is true? With the high number of species in the analysis, a detailed look would perhaps be beyond the scope of this paper, but an initial examination and preliminary description of some similarities along with some suggestion for future work based on this would be a nice addition to the manuscript.

Line 586: I suggest removing the word ‘Apparently’.

Line 588: I suggest changing ‘less false predictions’ to ‘fewer false predictions’

Line 591-593: Yes but it is also important to acknowledge here that with a much higher number of species, there is a much lower chance of correct classification by chance and more room for error. So, a similar overall classification success is actually a more successful model when there are more species. You have alluded to this earlier, but I think it is important to bring it up again here.

Line 595: I suggest using a word other than ‘significantly’ here.

Line 597: Should be past tense.

Line597 and 602: I think the _ should be a 5

Line 608: Much more closely than what?

Line 620: ‘focus’ should be ‘focused’

Lines 651-652: please provide a bit more detail here on the specific characteristics of insect sounds that may make them less ideal for the use of the mel-scale.

Lines 668-669: A minimum of 10 recordings per class seems like a reasonable minimum, but what about the duration of those recordings? Is just a few seconds of recordings enough? Please discuss this. Did the amount of data for a species affect classification success? How much data is necessary?

**Have the authors made all data and (if applicable) computational code underlying the findings in their manuscript fully available?**

Reviewer #1: **No: **

Reviewer #2: **No: **Currently, InsectSec32 is publicly available on Zenodo. The other datasets and Python code is not yet available.

Reviewer #3: Yes

PLOS authors have the option to publish the peer review history of their article (what does this mean?). If published, this will include your full peer review and any attached files.

Reviewer #1: **Yes: **Eamonn Keogh

Reviewer #2: **Yes: **Emmanuel Dufourq

Reviewer #3: No

Figure Files:

Data Requirements:

Reproducibility:

References:

---

## [Decision Letter · Decision Letter 1]

25 Sep 2023

Dear Dr Stowell,

We are pleased to inform you that your manuscript 'Adaptive representations of sound for automatic insect recognition' has been provisionally accepted for publication in PLOS Computational Biology.

Best regards,

Ricardo Martinez-Garcia

Academic Editor

PLOS Computational Biology

James O'Dwyer

Section Editor

PLOS Computational Biology

Reviewer's Responses to Questions

**Comments to the Authors:**

Reviewer #1: “Are the recordings that were not included in the datasets from these sources due to low quality of interest to the reviewers?”

Hmm, recall Diagoras’s story. He visited a temple on the Aegean island of Samothrace. Those who escaped from shipwrecks or were saved from drowning at sea would display portraits of themselves here in thanks to the great sea god Neptune. “Surely”, Diagoras was challenged by a believer, “these portraits are proof that the gods really do intervene in human affairs?” Diagoras' reply was “yea, but… where are they painted that are drowned?”

The “sources due to low quality” are similarly important negative data, they tell us what you could not work with.

“Note also that the mosquito classification of Batista et al is based on a trap device that records wingbeats optically: this successfully yields low-noise data, much lower noise than in general acoustic recordings. This helps to explain why the appropriate algorithm for their data, is not a fully general solution.” I guess. Maybe it is a argument for non-acoustic sensors ;-) Anyway, at least a dozen papers seem to have followed up, but use true audio data (including data from cell phones) and had success with Bayes.

I am not convinced by your arguments for deep learning over similar methods. In recent years there has been a bit of a push back against deep learning in time series forecasting, time series anomaly detection etc. However, I will not hold up your paper.

**Have the authors made all data and (if applicable) computational code underlying the findings in their manuscript fully available?**

Reviewer #1: Yes

PLOS authors have the option to publish the peer review history of their article (what does this mean?). If published, this will include your full peer review and any attached files.

Reviewer #1: **Yes: **Eamonn Keogh

---

## [Editor Report · Acceptance letter]

28 Sep 2023

PCOMPBIOL-D-23-00434R1 

Adaptive representations of sound for automatic insect recognition

Dear Dr Stowell,

I am pleased to inform you that your manuscript has been formally accepted for publication in PLOS Computational Biology. Your manuscript is now with our production department and you will be notified of the publication date in due course.

With kind regards,

Zsofia Freund
